# Kidney Toxicity of Drugs for the Heart: An Updated Perspective

**DOI:** 10.3390/metabo15030191

**Published:** 2025-03-11

**Authors:** Carlo Caiati, Roberto Arrigoni, Alessandro Stanca, Mario Erminio Lepera

**Affiliations:** 1Unit of Cardiovascular Diseases, Department of Interdisciplinary Medicine, University of Bari “Aldo Moro”, 70124 Bari, Italy; alessandrostanca@gmail.com (A.S.); marioerminio.lepera@uniba.it (M.E.L.); 2CNR Institute of Biomembranes, Bioenergetics and Molecular Biotechnologies (IBIOM), 70124 Bari, Italy; r.arrigoni@ibiom.cnr.it

**Keywords:** drugs for the heart, nephrotoxicity, glomerulopathies

## Abstract

Cardiovascular drugs are widely used for the prevention and treatment of various cardiac and vascular disorders. However, some of these drugs can also cause adverse effects on the kidney, leading to acute or chronic renal dysfunction, electrolyte imbalances, and increased mortality. The mechanisms of drug-induced renal toxicity vary depending on the type and class of the drug, the dose and duration of exposure, and the patient’s characteristics and comorbidities. In this review, we summarize the current knowledge on the renal effects of some common cardiovascular drugs, such as diuretics, angiotensin-converting enzyme inhibitors, angiotensin receptor blockers, calcium channel blockers, beta-blockers, antiplatelet agents, anticoagulants, and statins and proton-pump inhibitors. We also discuss the clinical implications and management strategies for preventing or minimizing drug-induced nephrotoxicity, as well as the potential role of oxidative stress in its pathogenesis.

## 1. Introduction

The kidney plays a vital role in maintaining fluid, electrolyte and acid–base balance homeostasis, regulating blood pressure, excreting metabolic waste products, producing hormones and vitamins, and finally regulating erythrocyte production. In particular, all waste products can reach a high concentration in the kidneys for three main reasons: (1) a very high renal perfusion, (2) the presence in renal tissues of a variety of xenobiotic transporters and metabolizing enzymes, and (3) the active reabsorption and concentration in the tubules of solutes during urine production [1]. Regarding the first point, the perfusion can be as high as from 800 to 1200 mL/min, which means almost 20% of the cardiac output. Since the total plasm is almost 3 L, in 24 h all the plasma is filtered 20 times. This high perfusion is driven by a very unique type of resistance that is arranged in parallel and not in series, so the total resistance is decreased. This accounts for the higher blood flow and high hydrostatic pressure in the glomerular capillary network. Then, the glomerular blood is connected by means of the efferent arteriole with a second capillary network: the peritubular vasa recta capillary network. This low pressure capillary network is the vascular highway for the immunocompetent cells moving to and from the interstitium and nephron tubular section [2]. Regarding the second point, the tubular cells are furnished with all the of xenobiotic transporter and metabolizing enzymes and are intermixed with the resident dendritic cells (macrophages) [3]. Therefore, every xenobiotic has a profound biochemical and also immunological contact with the nephrons and renal interstitium. Regarding the last point, since the proximal tubules reabsorb almost all the glomeral filtration in terms of Na^+^ and water, a high concentration of toxic substances/drugs is reached in the tubules and eventually in the interstitium [4].On the basis of this physiological background, the kidneys experience prolonged contact with drug molecules and become susceptible to damage by many drugs (even those not necessarily targeting the kidney). Therefore, drugs can alter kidney structure and function [5].

Drug-induced nephrotoxicity is a common problem in clinical medicine and the incidence of drug-related acute kidney injury (AKI) may be as high as 60 percent [6]. The condition can be costly and may require multiple interventions, including hospitalization. Drug-induced nephrotoxicity can also lead to chronic kidney disease (CKD), end-stage renal disease (ESRD), and increased cardiovascular morbidity and mortality [7].

Cardiovascular drugs are among the most frequently prescribed medications worldwide [8], as they are used for the prevention and treatment of various cardiac and vascular disorders, such as hypertension, heart failure, ischemic heart disease, arrhythmias, stroke, and peripheral arterial disease. However, some of these drugs can also cause adverse effects on the kidney, either directly or indirectly. The mechanisms of drug-induced renal toxicity vary depending on the type and class of the drug, the dose and duration of exposure, and the patient’s characteristics and comorbidities [9].

In this review, we summarize the current knowledge on the renal effects of some common cardiovascular drugs, such as diuretics, angiotensin-converting enzyme inhibitors (ACEIs), angiotensin receptor blockers (ARBs), calcium channel blockers (CCBs), beta-blockers (BBs), antiplatelet agents (APAs), anticoagulants (ACs), and statins and proton-pump inhibitors (PPIs). We also discuss the clinical implications and management strategies for preventing or minimizing drug-induced renal toxicity. In Figure 1, the structures of each class of cardiovascular drugs are reported.

## 2. Mechanisms of Drug-Induced Nephrotoxicity

The mechanisms of drug-induced nephrotoxicity may differ between various drugs or drug classes, and they are generally categorized based on the histological component of the kidney that is affected. Some common factors that damage the kidneys are changes in glomerular hemodynamics, tubular cell toxicity, inflammation, crystal nephropathy, rhabdomyolysis, and thrombotic microangiopathy [10].

### 2.1. Changes in Glomerular Hemodynamics

The glomerular filtration rate (GFR) of healthy young individuals is 120 mL per minute [11]. The kidneys can regulate the blood flow in the afferent and efferent arterioles to keep a constant filtration rate and to ensure the proper urine output. Filtration takes place in the glomerulus, which is made up of a network of highly permeable capillaries that filter blood and form urine. This filtration is regulated by glomerular hemodynamics, which refers to the blood flow and pressure within the glomerulus. Changes in glomerular hemodynamics can affect the filtration rate and the permeability of the glomerular membrane, leading to proteinuria, hematuria, or reduced renal function [12]. Some drugs can alter glomerular hemodynamics by affecting the balance between vasoconstrictors and vasodilators, such as angiotensin II, prostaglandins, nitric oxide, and endothelin. For example, nonsteroidal anti-inflammatory drugs (NSAIDs) can inhibit prostaglandin synthesis and cause afferent arteriole constriction, reducing renal blood flow and glomerular filtration rate [13]. Similarly, angiotensin-converting enzyme inhibitors (ACEIs) and angiotensin receptor blockers (ARBs) can block angiotensin II-mediated efferent arteriole constriction and cause glomerular capillary pressure to drop, resulting in a decreased filtration fraction [14]. Other drugs that can cause changes in glomerular hemodynamics include calcineurin inhibitors (cyclosporine and tacrolimus), contrast media, amphotericin B, cisplatin, and gentamicin [9].

### 2.2. Tubular Cell Toxicity

Tubular cells are responsible for reabsorbing water and solutes from the filtrate and secreting waste products into the urine. They also express various transporters, enzymes, and receptors that are involved in drug uptake, metabolism, and excretion. The renal proximal tubules are highly susceptible to drug toxicity, as these tubules are maximally exposed to the drugs that undergo maximal reabsorption and concentration in that part of the nephron [15]. Some drugs can directly or indirectly impair the function or integrity of tubular cells, leading to acute tubular necrosis, tubulo-interstitial inflammation, or obstructive nephropathy. Drug-induced cytotoxicity results from the damage to the mitochondria in the tubules, the disruption of the tubular transport system, and the increase in oxidative stress due to free radical production [16]. Some of the most well-known drugs that exert drug-related tubular toxicity are aminoglycosides, amphotericin, and some antivirals. They disrupt the cell membrane of tubular cells and also the mitochondria organelles. In particular, aminoglycosides can bind to phospholipids in the cell membrane and inhibit protein synthesis, resulting in cell swelling, vacuolization, and apoptosis [17]. Amphotericin B can disrupt the cell membrane and increase its permeability to cations, causing cell dysfunction and necrosis [18,19]. Antivirals such as adefovir and foscarnet can inhibit mitochondrial DNA polymerase and cause mitochondrial toxicity [20,21]. Other drugs that can cause tubular cell toxicity include cisplatin, methotrexate, acyclovir, vancomycin, and contrast media [6,22].

### 2.3. Inflammation

Many drugs that are nephrotoxic can cause inflammation with activation of the immune system in the glomerulus, the proximal tubules, and the surrounding extracellular matrix; this can lead to fibrosis of the kidney tissue [23]. Some drugs can induce inflammation by triggering a hypersensitivity reaction that involves T cells, antibodies, or complement activation. For example, penicillins, cephalosporins, sulfonamides, and rifampin can cause acute interstitial nephritis (AIN), which is characterized by fever, rash, eosinophilia, and hematuria [24]. NSAIDs can also cause interstitial nephritis by inhibiting prostaglandin synthesis and increasing leukotriene production [25]. Chronic interstitial nephritis occurs frequently with long-term use of calcineurin inhibitors, lithium, some anticancer drugs, or analgesics [26,27,28,29]. In case of chronic interstitial nephritis, early detection is especially important because it is difficult to diagnose until most of the kidney function is lost.

### 2.4. Crystal Nephropathy

Crystal nephropathy refers to the formation and deposition of insoluble crystals within the renal tubules, causing obstruction, inflammation, and injury. Crystal nephropathy can occur as a result of endogenous or exogenous factors, such as hyperuricemia, hyperoxaluria, or drug administration [30]. The risk factors for developing crystal nephropathy include dehydration, high dose or prolonged use of the drug, renal impairment, and metabolic acidosis [31]. Some drugs can form crystals in the urine due to their low solubility, high dose, or acidic pH. For example, acyclovir can precipitate as needle-shaped crystals in the distal tubules, especially when given as rapid intravenous bolus or in dehydrated patients [32,33]. Sulfonamides can form crystals that are associated with casts and interstitial nephritis [34,35]. Methotrexate can cause intratubular precipitation of 7-hydroxymethotrexate, which is a metabolite with low solubility [36]. Other drugs that can cause crystal nephropathy include indinavir, triamterene, atazanavir, and ciprofloxacin [37].

### 2.5. Rhabdomyolysis

Rhabdomyolysis is a syndrome of skeletal muscle breakdown that releases intracellular contents, such as myoglobin, creatine kinase, and electrolytes, into the circulation. Rhabdomyolysis can cause AKI by several mechanisms, such as the direct tubular toxicity of myoglobin, tubular obstruction by myoglobin casts, vasoconstriction and ischemia, oxidative stress, and inflammation [38]. Rhabdomyolysis can be triggered by various factors, such as trauma, ischemia, infection, exercise, or drugs. Some drugs can cause rhabdomyolysis directly by inducing muscle damage or indirectly by enhancing the effects of other factors. For example, statins can cause rhabdomyolysis by inhibiting the synthesis of coenzyme Q10 and impairing mitochondrial function [39,40]. Cocaine can cause rhabdomyolysis by inducing vasoconstriction, hyperthermia, and seizures [41]. Colchicine can cause rhabdomyolysis by disrupting microtubule function and inhibiting cellular transport [42]. Other drugs that can cause rhabdomyolysis include antipsychotics, antidepressants, antihistamines, antimalarials, and opioids [43,44,45,46].

### 2.6. Thrombotic Microangiopathy

Thrombotic microangiopathy (TMA) is a disorder characterized by microvascular thrombosis, hemolytic anemia, thrombocytopenia, and organ damage, especially in the kidneys. TMA can be caused by various conditions, such as genetic defects, infections, autoimmune diseases, or drugs [47]. Drug-induced TMA (DITMA) can be classified into immune-mediated and dose-related/toxic mechanisms, depending on the type of drug and the timing of onset. Immune-mediated DITMA involves drug-dependent antibodies that activate platelets or endothelial cells, leading to thrombosis and inflammation. Dose-related/toxic DITMA involves drugs that directly damage the endothelium or interfere with its function, leading to the exposure of the subendothelial matrix and the activation of coagulation [48]. For example, clopidogrel and ticlopidine can cause TMA by inducing antibodies against platelet glycoproteins or endothelial cells [49]. Cyclosporine and mitomycin-C can cause TMA by damaging the endothelium and activating the coagulation cascade [50]. Quinine can cause TMA by binding to platelets and inducing complement activation [51]. Other drugs that can cause TMA include gemcitabine, bevacizumab, and sunitinib [48,52,53,54].

## 3. Renal Toxicity of Cardiovascular Drugs

Figure 2 provides a brief summary of the mechanisms and sites of renal damage caused by cardiovascular drugs.

### 3.1. Diuretics

Diuretics are drugs that increase urine output by inhibiting sodium reabsorption in different segments of the nephron. They are widely used for the treatment of hypertension, heart failure, edema, and other conditions associated with fluid overload. Diuretics can be classified into different groups according to their site of action: loop diuretics (e.g., furosemide), thiazide diuretics (e.g., hydrochlorothiazide), potassium-sparing diuretics (e.g., spironolactone), osmotic diuretics (e.g., mannitol), and carbonic anhydrase inhibitors (e.g., acetazolamide) [55].

Loop diuretics, in particular, are drugs that increase the elimination of water and electrolytes by blocking the sodium/potassium/chloride cotransporter in the thick ascending limb of Henle’s loop. They are often used in patients who have or are at risk of acute kidney injury (AKI) for various indications, such as volume overload, hyperkalemia, hypercalcemia, and hyperazotemia. They may have important benefits in critically ill patients who receive large volumes of fluids and can prevent or treat fluid overload and pulmonary edema, which can improve oxygenation and hemodynamics [56]. However, they can also have negative effects on renal function and outcomes in AKI patients.

Loop diuretics can cause a reduction in the effective circulating volume by inducing venodilation or diuresis, which can lead to a reduction in renal blood flow and glomerular filtration rate through the stimulation of the renin–angiotensin–aldosterone system (Figure 2) [57]. Although some studies suggest that loop diuretics can prevent tubular obstruction by increasing urine flow and flushing out tubular debris, other studies suggest that loop diuretics can worsen tubular obstruction by acidifying the urine and enhancing the aggregation of Tamm–Horsfall protein in the tubules [58]. Loop diuretics can also cause electrolyte disturbances and metabolic alkalosis [59]. High doses of loop diuretics can be ototoxic and cause hearing impairment or ringing in the ears [60]. Loop diuretics may also affect mucociliary clearance in the respiratory tract and have some immune-suppressive effects [61].

Diuretics are usually used in CHF to relieve the symptoms of pulmonary edema. Many clinical trials have suggested that those patients with heart failure who developed worsening renal failure had a higher central venous pressure (CVP) or intra-abdominal pressure (often caused by ascites or edema of internal organs), so that treatments of reducing CVP and intra-abdominal pressure (mostly using diuretics) were given to CHF patients to prevent worsening renal failure, so as to reduce mortality [62,63,64]. However, growing evidence shows that heavy dependence on the strategy of diuretics to achieve this goal may adversely affect renal function and outcome [65]. Moreover, loop diuretics block sodium chloride uptake in the macula densa, independent of any effect on sodium and water balance, thereby stimulating the RAAS, and leading to AKI. Recently, some trials have suggested that excessive diuresis is harmful since it worsens renal function in HF patients, which increases mortality [66,67].

Sometimes, AKI was caused by the combination of diuretics and other agents, such as antibiotics, ACEIs/ARBs, and NSAIDs. Diuretics are often used in combination with ACE inhibitors to control blood pressure, but ACEIs, by causing efferent arteriolar dilatation, can further reduce intraglomerular pressure, eventually leading to AKI [68] (Figure 2). Therefore, as clearly emerges from the literature [68], caution should generally be exercised when using diuretics in association with ACE inhibitors, especially in those patients with underlying renal disease.

NSAIDs could inhibit PGI2 synthesis so as to affect intraglomerular hemodynamics, and this is riskier in causing AKI when combined with diuretics [25].

Despite their widespread use in AKI, there is no clear evidence that loop diuretics improve outcomes in AKI. Loop diuretics do not decrease mortality, dialysis requirements, or ICU length of stay in AKI patients. Therefore, the use of loop diuretics should not be based solely on urine output, but on careful assessment of volume status, renal function, and electrolyte balance [58].

Finally, it is useful to mention a rare but serious complication of treatments with triamterene. Triamterene is a potassium-sparing diuretic used in combination with thiazide diuretics which is frequently associated with crystalluria, even if it can rarely cause a serious nosological entity defined as crystalline nephropathy [69]. Crystalline nephropathy is a type of kidney disease characterized by the histologic finding of crystal deposition within the kidney parenchyma, which can lead to renal tubular injury, inflammation, fibrosis, and dysfunction [30]. The best way to prevent crystalline nephropathy is to check serum creatinine and urine sediment regularly, avoid volume depletion, nonsteroidal anti-inflammatory drug use, and acid urine in patients who are at high risk. Urine alkalinization may help to prevent urinary crystal and cast formation; however, the treatment also involves stopping triamterene and alkalinizing urine in patients who do not have oliguria [30].

### 3.2. Angiotensin-Converting Enzyme Inhibitors and Angiotensin Receptor Blockers

ACEIs and ARBs are drugs that inhibit the renin–angiotensin–aldosterone system (RAAS), a hormonal system that regulates blood pressure, fluid and electrolyte balance, and vascular tone. ACEIs block the conversion of angiotensin I to angiotensin II, a potent vasoconstrictor and pro-inflammatory peptide, while ARBs block the binding of angiotensin II to its receptors. ACEIs and ARBs are widely used for the treatment of hypertension, heart failure, diabetic nephropathy, and other cardiovascular and renal diseases. They have been shown to reduce cardiovascular morbidity and mortality, as well as to slow the progression of CKD and ESRD [70,71].

ACE inhibitor therapy can often improve renal blood flow (RBF) and sodium excretion rates in congestive heart failure (CHF) and slow down the progression of chronic renal disease, but it can also cause a syndrome of “functional renal insufficiency” and/or hyperkalemia. This type of AKI usually occurs soon after starting ACE inhibitor therapy but can also happen after months or years of therapy, even without any previous adverse effects [72,73,74].

Various mechanisms are implicated in the development of AKI in patients undergoing ACE inhibitor therapy. One primary cause is a reduction in renal perfusion due to a fall in mean arterial pressure (MAP) that is insufficient to maintain adequate renal perfusion or that triggers a significant reflex activation of the renal sympathetic nerves [75]. Additionally, ACE inhibitor therapy can lead to hypotension through other potential mechanisms, such as an increase in vasodilatory prostaglandins or a decrease in total peripheral resistance, particularly in cases where cardiac output remains unchanged due to heart failure [76,77,78]. Another mechanism is in patients who are volume depleted due to diuretic therapy. In this case, ACE inhibitors can cause ARF in patients who have CHF and are undergoing diuretic therapy. Studies have shown that patients undergoing diuretic therapy who are given ACE inhibitors have a higher risk of developing AKI compared with those who are not taking diuretics [79,80]. The reason for this is that diuretics cause a decrease in blood volume, leading to reduced blood flow to the kidneys. ACE inhibitors further reduce blood flow to the kidneys by causing vasodilation of the efferent arterioles, which can result in a decrease in GFR and an increase in serum creatinine levels [81]. Another condition includes patients with high-grade bilateral renal artery stenosis or patients with atherosclerotic disease in smaller preglomerular vessels or with afferent arteriolar narrowing due to hypertension [72,82,83,84]. Moreover, ACE inhibitors may precipitate AKI in patients taking vasoconstrictor agents, such as nonsteroidal anti-inflammatory agents. This is because these agents can cause vasoconstriction of the afferent arteriole, which reduces the renal perfusion and exacerbates the previously well-described effects of diuretics and ACE inhibitors. This combination has been known as the “Triple whammy” and can result in a significant reduction of the GFR and glomerular perfusion [85,86,87].

The management of AKI from ACE inhibitors involves prompt recognition, discontinuation of the offending drug, and a close follow-up for patients with a higher risk such as those with chronic kidney disease, heart failure, or volume depletion. Therefore, these patients should be monitored closely for changes in serum creatinine and potassium levels when starting or adjusting ACE inhibitor therapy. If AKI occurs, the ACE inhibitor should be stopped immediately and the patient should be assessed for volume status, blood pressure, electrolytes, and urine output. Fluid resuscitation may be needed to restore renal perfusion pressure and improve GFR. However, excessive fluid administration should be avoided in patients with heart failure or pulmonary edema [88]. In some cases, vasopressors may be required to maintain adequate blood pressure [89]. Hyperkalemia may occur as a result of reduced potassium excretion and should be treated with appropriate measures, such as sodium bicarbonate, insulin and glucose, calcium gluconate, or potassium binders [90]. Dialysis may be necessary in severe cases of AKI or hyperkalemia that are refractory to medical therapy [91].

### 3.3. Calcium Channel Blockers

CCBs are drugs that inhibit the influx of calcium ions into vascular smooth muscle cells and cardiac myocytes, resulting in vasodilation and negative inotropic and chronotropic effects.

Numerous studies have demonstrated that CCBs are less effective than other medications in reducing proteinuria and preventing kidney damage in patients with proteinuric kidney diseases, regardless of whether they have diabetes or not. This is due to the fact that CCBs can compromise the kidney’s ability to regulate its own blood pressure, resulting in increased pressure in the glomeruli [92]. CCBs may cause, in fact, glomerular autoregulation impairment by dilating the afferent glomerular artery, thereby eliminating the potent mechanism that shields the glomerular capillaries from systemic pressure transmission. This effect can lead to increased proteinuria and potentially progressive damage to the glomerular capillary network, especially when a hypertensive condition persists [93]. This clarifies why CCBs have been found to be less effective at safeguarding the kidneys in clinical trials than RAS blockers, not because RAS blockers have additional advantages [94].

Moreover, some CCBs may also interact with other drugs metabolized by the cytochrome P450 3A4 enzyme, such as clarithromycin, and increase the risk of acute kidney injury due to hypotension and hypoperfusion [95]. The risk is higher for nifedipine, felodipine, and amlodipine, and lower for verapamil and diltiazem [96]. Therefore, possible approaches could involve discontinuing the use of the calcium channel blocker for the duration of clarithromycin treatment or selecting an antibiotic that does not inhibit CYP3A4 when it is clinically appropriate.

### 3.4. Beta-Blockers

BBs are drugs that inhibit the binding of catecholamines to beta-adrenergic receptors in the heart, blood vessels, and other tissues, resulting in negative inotropic and chronotropic effects, vasodilation, and reduced renin secretion. BBs are widely used for the treatment of hypertension, angina pectoris, arrhythmias, heart failure, and other cardiovascular disorders. However, BBs may also have an impact on renal function, as alpha-, beta 1-, and beta 2-adrenergic receptors in the kidney mediate vasoconstriction, renin secretion, and vasodilatation, respectively [97].

The effects of beta-blockers on renal function may depend on several factors, such as the degree of cardioselectivity (i.e., the selectivity for beta-1 receptors over beta-2 receptors), the intrinsic sympathomimetic activity (i.e., the partial agonist activity at beta receptors), the lipid solubility (i.e., the ability to cross the blood–brain barrier), and the route of administration (i.e., oral or intravenous) of the drug [97]. The administration of beta-blockers, regardless of whether they are cardio-selective or have intrinsic sympathomimetic activity, typically leads to a decrease in GFR [98,99,100]. This could be explained by several mechanisms such as by lowering cardiac output, blocking beta 2-receptors, or stimulating alpha-receptors [101]. However, it is widely known that sympathetic over-activity is a component of CKD and plays a fundamental role in sustaining hypertension and the resulting cardiac complications [102]. Even though the administration of BBs can result in statistically significant changes in renal function, these changes are typically not deemed clinically significant, even in patients who have pre-existing renal disease [97,103]. For these reasons, it is unfortunate that β-blockers are not being utilized to their maximum potential due to apprehensions regarding their possible adverse effects on renal function and glycemic control, although it is important to remember that some BBs, such as atenolol, nadolol, and sotalol, are not metabolized and are excreted by the kidney, so their dose must be adjusted according to the level of renal function [104].

### 3.5. Antiplatelet Agents

APAs are drugs that interfere with platelet activation, clumping, and clot formation. Aspirin is the most commonly used antiplatelet drug, and it permanently inhibits cyclooxygenase (COX), which, at low doses, is used to treat or prevent cardiovascular problems and reduce the risk of cardiovascular disease. P2Y12 receptor blockers, including clopidogrel, prasugrel, and ticagrelor, are mainly used in combination with aspirin to treat acute coronary syndrome and prevent clotting in stents after percutaneous coronary intervention. However, these drugs can also cause kidney damage, especially in patients with chronic kidney disease [105].

Low-dose aspirin, with a dosage of 100 mg, is classified as a nonsteroidal anti-inflammatory drug (NSAID). Historically, NSAIDs have been considered unsafe for use in patients with chronic kidney disease (CKD) due to several mechanisms [106]. Firstly, aspirin and NSAIDs are drugs that block the production of prostaglandins, which are lipid mediators that regulate various physiological processes, including inflammation, pain, and blood flow [107]. Prostaglandins are synthesized from arachidonic acid by the enzyme COX, which has two isoforms: COX-1 and COX-2. COX-1 is constitutively expressed in most tissues and is responsible for the production of prostaglandins that protect the gastric mucosa, regulate platelet aggregation, and maintain renal perfusion. COX-2 is inducible by inflammatory stimuli and is responsible for the production of prostaglandins that mediate inflammation, pain, and fever [108]. Aspirin and NSAIDs inhibit both COX-1 and COX-2, but with different degrees of selectivity. Aspirin irreversibly acetylates COX-1 and COX-2, while most NSAIDs reversibly bind to the active site of both isoforms [109]. By inhibiting COX-1 and COX-2, aspirin and NSAIDs reduce inflammation and pain, but they also cause side effects in the gastrointestinal tract, kidney, and platelets [110]. In the kidneys, prostaglandins act as vasodilators of the afferent arteriole, increasing renal blood flow and GFR, but they can also modulate the activity of the renin–angiotensin–aldosterone system and the sympathetic nervous system to ensure sufficient blood flow to the organs [111,112]. Prostaglandins also regulate sodium and water excretion by influencing tubular reabsorption and have an antagonistic effect on the receptors for antidiuretic hormone (ADH), thereby promoting diuresis [113,114]. By blocking the production of prostaglandins, aspirin and NSAIDs impair these renal functions, resulting in a reduction of total renal perfusion and GFR and sodium and water retention, leading to renal vasoconstriction and medullary ischemia and finally culminating in AKI or, in the long term, worsening CKD [25,115,116]. The risk of aspirin- and NSAID-induced kidney damage is higher in patients who have dehydration, heart failure, liver cirrhosis, sepsis, or use of other nephrotoxic drugs, such as diuretics and ACEIs [117,118].

Hence, the main form of AKI caused by aspirin and NSAIDs is hemodynamically mediated, due to reduced renal perfusion and ischemia. However, there exists a second form of AKI caused by aspirin and NSAIDs which is AIN (Figure 2). AIN is based on a delayed hypersensitivity reaction to these drugs that causes inflammation and edema of the renal interstitium, which leads to nephrotic proteinuria or acute tubular necrosis a few days after the initiation of treatment, which is typically restored after discontinuation of the drug [119,120,121,122]. Instead, long-term use of aspirin and NSAIDs can lead to CKD by causing chronic interstitial nephritis or papillary necrosis [25,123].

The prevention of aspirin and NSAID-induced kidney damage involves the careful selection and monitoring of these drugs in patients who are at high risk of renal injury, especially those with advanced age and pre-existing renal disease. The lowest effective dose and shortest duration of treatment should be used, and alternative analgesic or anti-inflammatory agents should be considered if possible.

Although there is limited evidence that P2Y12 inhibitors can cause renal damage, some studies have suggested that these drugs may be potentially hazardous, particularly when combined with other nephrotoxic medications such as statins [93]. For instance, the combination of ticagrelor and rosuvastatin is commonly used in patients with acute coronary syndrome and has been associated with AKI in some studies [124,125]. The underlying mechanism of this association is unknown and is not likely to be related to hepatic metabolism interactions, given that ticagrelor is mainly metabolized by CYP3A4 while rosuvastatin is primarily metabolized by CYP2C9 [124]. It is possible that ticagrelor may cause a transient worsening of renal function through an unknown mechanism, thereby enhancing the ability of rosuvastatin to induce rhabdomyolysis and subsequent AKI. Therefore, caution should be exercised when combining P2Y12 inhibitors with other nephrotoxic drugs, and close monitoring of renal function is recommended in patients receiving such combination therapy.

Finally, some studies have shown that clopidogrel may be a cause of thrombotic microangiopathy, a condition characterized by renal failure, hemolytic anemia, and thrombocytopenia [126,127]. Due to the widespread use of clopidogrel, clinicians should exercise caution and remain vigilant regarding this infrequent but potentially serious complication.

### 3.6. Anticoagulants

ACs are drugs that inhibit the coagulation cascade, which is responsible for blood clotting and hemostasis. They are used to prevent or treat thromboembolic disorders, such as deep vein thrombosis, pulmonary embolism, atrial fibrillation, or stroke. There are different types of anticoagulants, such as vitamin K antagonists (e.g., warfarin), direct thrombin inhibitors (e.g., dabigatran), and direct factor Xa inhibitors (e.g., rivaroxaban, apixaban). Studies have demonstrated the involvement of these drugs in a new nosological entity called anticoagulant-related nephropathy (ARN).

ARN, previously known as warfarin nephropathy, is a relatively recently described entity despite the fact that warfarin has been used for several decades. ARN is a form of AKI that can occur in patients with a supratherapeutic international normalized ratio (INR), as well as those taking newer anticoagulants [128,129]. When there is no apparent cause of AKI, and the patient has recently had a supratherapeutic INR and hematuria, ARN may be considered as a possible diagnosis. Renal biopsy can help to confirm the presence of ARN in such cases; the absence of active inflammatory lesions and the presence of RBCs and RBC occlusive casts tubules and Bowman space are indicative of ARN [130]. Recent research has tried to shed light on the complex and multifactorial mechanism underlying ARN. One of the key factors is a reduction in the number of functional nephrons, which can lead to the over-proliferation of the surviving glomeruli and glomerular hypertension, rendering them vulnerable to glomerular hemorrhage [131]. It has been proposed that the combination of mild glomerular disease and ARN may lead to glomerular hematuria and a significant accumulation of RBCs within nephrons. If urinary flow is diminished due to interstitial inflammation or changes in blood pressure and most of all dehydration, intratubular RBCs may form occlusive casts, leading to the development of AKI [132]. This could explain why patients with pre-existing chronic kidney disease are at higher risk of developing this condition. In addition, studies have described intricate interactions involving these molecules, which can trigger a cascade of oxidative stress and inflammation in the renal tubular epithelium and surrounding interstitium [133]. Finally, recent studies have proposed alternative pathways involving a decrease in protein C levels and abnormal signaling of endothelial protein C receptors [134].

Therefore, when a patient presents with unexplained AKI that does not resolve, ARN should be considered, and a renal biopsy should be proposed to confirm the diagnosis. If ARN is diagnosed, a switch from warfarin to DOACs should be considered, or a reduction in DOAC dosage if the patient is already on DOACs [130]. However, further studies are needed to establish guidelines for the management of ARN, as current guidelines are lacking. Additionally, close monitoring of kidney function and anticoagulant therapy is essential for the management of these patients.

### 3.7. Statins

Statins, also known as 3-hydroxy-3-methylglutaryl coenzyme A (HMG-CoA) reductase inhibitors, are a commonly prescribed class of drugs used for the management and treatment of hypercholesteremia.

They work by reducing the levels of low-density lipoproteins (LDL), total cholesterol, and triglycerides, and at the same time, in a modest unpredictable way, by increasing high-density lipoproteins (HDL); the effect of statins on cholesterol along with a cholesterol independent “pleiotropic” effect translates into a potential antiatherogenic effect [135]. As such, they are widely used for both the primary and secondary prevention of coronary heart disease. Statins, however, can cause numerous unwanted side effects that can cause discontinuation of the drug. Guidelines on cholesterol and statins are often created by experts who have conflicts of interest so we think that the officially reported data on statins’ toxicity are downplayed [136]. The main reported collateral effects are statin-associated muscle symptoms (SAMSs) such as myalgia, myopathy, myositis, and even worse rhabdomyolysis, as well as new onset type 2 diabetes mellitus, neuropsychiatric effects such as depression, sleep problems etc., hepatotoxicity, microbiome-mediated effects, and finally renal toxicity, to name only some [137].

Statin-induced kidney injury is a rare but serious complication of statin therapy. The statins that have a more direct impact on the kidneys are the hydrophilic ones such as pravastatin and rosuvastatin, especially at high doses [136,138]. However, even the less hydrophilic ones such as simvastatin and atorvastatin, which are more extensively metabolized by the liver with minimal clearing by the kidneys, can exert significant nephrotoxicity [93,139,140,141]. Statin-induced nephrotoxicity can involve several mechanisms. One possible mechanism involves the potential of statins, especially at high dosages, to cause rhabdomyolysis; this is a condition characterized by the breakdown of skeletal muscle, which can lead to the release of sarcoplasmic protein and electrolytes in the blood. The sarcoplasmic protein can then be filtered in the glomerulus and accumulates in the tubules, thus causing acute renal failure and other life-threatening complications, such as hyperkalemia and cardiac arrhythmias [142,143]. Another less-known but more clinically relevant mechanism is acute or subacute tubulo-interstitial nephritis, provoked by statins at high dosage, which can lead to a direct tubular damage that is related to the cumulative metabolic effects of the statin on tubular cells [139] (Figure 2). Last but not least, there is a growing body of evidence suggesting that statin toxicity may be closely linked to oxidative stress as it can be deduced by the numerous functional reactive groups in the molecule (Figure 1). It has been found that statins can generate reactive oxygen species (ROS) during metabolism, leading to oxidative stress and varying levels of toxicity, including damage to skeletal muscle, the liver, and kidneys [144]. Moreover, statin-induced suppression of coenzyme Q10 (CoQ-10) production, which has antioxidant properties, has been proposed as another potential cause of AKI [145]. CoQ-10 is an enzyme that plays a role in generating adenosine triphosphate (ATP) in the mitochondria, and its suppression can impair the mitochondrial respiratory chain, leading to mitochondrial dysfunction [146]. This effect could also be supported by the ability of statins to directly inhibit complexes I and III and to trigger mitochondria-induced calcium signaling alteration, enhancing ROS and promoting inflammation and apoptosis [144,147,148]. Overall, statin-induced tubulo-interstitial nephritis is probably underreported because it evolves insidiously in patients who are prone to develop AKI for other reasons (e.g., comorbid conditions such as diabetes and arterial hypertension, concomitant nephrotoxic drug treatment, etc.) [93,149].

Identifying statin-induced kidney injury is crucial, and physicians should exercise caution in dosing statins. Patients using high doses of statins should be educated about the potential side effects. Although clinical rhabdomyolysis with statins is rare, it should be considered in patients who present with weakness and renal failure after a cardiac intervention [150]. A thorough drug history analysis can aid in the early detection and treatment of statins’ adverse effects. Treatment typically involves stopping the medication and potentially switching to an alternative regimen. Apart from rhabdomyolysis, the renal toxic effects of statins, in particular tubular interstitial damage, should be carefully monitored. The best way to pursue this aim is by evaluating for progressive augmentation of protein in the urine, essentially by searching abnormal casts only detectable on direct urine microscopy by an experienced observer. In case such protein elevation is documented, statin suspension is warranted; you do not have to wait for the decline of kidney filtration function (the reduction of creatine clearance) that indicates already irreversible and conspicuous statin-driven damage [93,139].

### 3.8. Proton-Pump Inhibitors

PPIs are a class of drugs that are commonly used to treat acid-related disorders such as peptic ulcers, gastroesophageal reflux disease, and *Helicobacter pylori* infection by reducing the production of gastric acid through inhibition of the enzyme H^+^/K^+^ATPase in the stomach. Although not primarily cardiovascular drugs, PPIs are also widely used in cardiology as gastroprotective agents for patients taking antiplatelet and anticoagulant medications, as these medications increase the risk of gastrointestinal bleeding. However, PPIs have also been associated with several adverse effects, including AIN, AKI, and CKD [151]. An immune-mediated reaction to the deposition of PPIs or their metabolites in the tubulo-interstitium is responsible for the formation of an interstitial inflammatory infiltrate and edema, which leads to the development of AIN and AKI. This results in acute inflammation and tubulo-interstitial damage, which can ultimately progress to interstitial fibrosis and chronic interstitial nephritis that, if left untreated, can lead to CKD and, in severe cases, even renal failure [152]. Another potential side effect associated with the use of PPIs is hypomagnesemia, which may affect kidney function by dysregulating vascular and endothelial function, contributing to the progression of CKD [153,154]. Moreover, numerous studies have shown that the use of PPIs is linked to a higher risk of enteric infections, such as *C. difficile* infection, which, in turn, can lead to dehydration-associated AKI [155,156]. Finally, studies suggest that mitochondrial injury may play a key role in inducing necrotic cell death in proximal tubular cells, and the promotion of such necrotic cell death by PPIs was found to be facilitated by ROS [157]. The necrosis induced by PPIs may lead to the release of cell debris, which could trigger an immune response and contribute to the development of acute tubulo-interstitial nephritis observed in PPI-treated patients [158]. Additionally, the risk of PPI-induced nephrotoxicity may be increased by additional environmental factors, including comorbidities and concomitant medications such as oral anticoagulants that can lead to iron overload in proximal tubular cells. Iron overload was found to facilitate PPI-induced nephrotoxicity in cultured cells. The combined effect of these factors may explain why GFR is lost at a faster rate in patients on chronic PPI therapy [157].

Considering the medical relevance of PPIs, it is crucial to ensure that these medications are used properly in accordance with therapeutic guidelines. Close monitoring of the benefits derived from PPI use is necessary, and discontinuation (with gradual tapering) of the drug therapy should occur promptly once it is no longer required. As part of routine clinical practice, monitoring of GFR on monthly basis may be helpful in detecting the potential adverse effects of PPI use.

### 3.9. Contrast Media

Although contrast media (CM) is not strictly a cardiovascular drug, it has an increasing importance in the context of cardiac procedures, such as coronary angiography, percutaneous coronary intervention, and the escalating use of coronary computed tomography [159], where it is used to enhance the visualization of vascular structures.

Contrast-induced nephropathy (CIN) has been defined as the occurrence of acute renal impairment within 2–7 days after iodinated contrast media (CM) administration [160]. It is a serious adverse effect that can lead to acute AKI and increased morbidity and mortality. Several mechanisms interact in a complex manner to contribute to the pathophysiology of CIN [161]. After an initial phase of vasodilatation, contrast media (CM) triggers intense vasoconstriction due to the inhibition of nitric oxide-mediated vasodilatation, changes in intracellular calcium concentration, and the release of adenosine and endothelin [162,163,164]. Then, the major damaging renal mechanism comes into play: the massive oxidative interaction of CM with renal tubular cells and endothelial cells with the consequent release of ROS. However, according to Li et al., the increase in ROS synthesis appears to be more a consequence of direct CM toxicity on tubular cells than the cause of cellular damage. This process is favored by the sustained reduction in renal blood flow with the consequent ischemia even of the outer regions of the medulla, resulting finally in acute tubular necrosis [165]. The route of administration and the chemical properties of CM play a crucial role in the development of CIN, with intra-arterial administration being more nephrotoxic than intravenous injection [161,166]. Other factors that contribute to the development of AKI include hypotension, microembolization of atheromatous debris, or bleeding complications, leading to ischemic acute tubular necrosis [167]. The risk factors for CIN can be exacerbated by hemodynamic alterations and medications, such as metformin, which can lead to lactic acidosis in the presence of kidney dysfunction and AKI [168,169].

The prevention of CIN is crucial as there is currently no targeted treatment available. Some common general measures that can be taken to prevent CIN include limiting the volume of CM and discontinuing the use of nephrotoxic drugs at least 48 h before exposure to CM [161]. Moreover, hydration with intravenous fluids is recommended before and after the procedure, with a sliding scale protocol based on left ventricular end-diastolic pressure [168]. Bicarbonate infusion can also help to prevent renal tubular fluid injury by alkalinizing the environment and scavenging ROS [168].

A brief summary of nephrotoxic effect of cardiovascular drugs is reported in Table 1.

## 4. Role of Oxidative Stress in Drug-Induced Nephrotoxicity

Oxidative stress is a condition of imbalance between the production and elimination of ROS, and/or decreased antioxidant defense activity, which can cause damage to cellular components such as lipids, proteins, and DNA (Figure 2) [170]. Oxidative stress is involved in the pathogenesis of various diseases, including drug-induced nephrotoxicity. Several drugs can induce oxidative stress in the kidney by different mechanisms, such as increasing ROS generation, decreasing antioxidant defense, or impairing mitochondrial function [171]. Oxidative stress can contribute to renal injury by activating inflammatory responses, inducing apoptosis, modulating redox signaling, and altering gene expression [172].

It is widely recognized that CKD is characterized by increased levels of oxidative stress [173]. The kidneys are highly susceptible to damage caused by oxidative stress due to the intense oxidative activity of their mitochondria. CKD is characterized by increased levels of oxidative stress, which result from both a depletion of antioxidants and the over-production of ROS. It has been demonstrated that this excessive ROS generation leads to the oxidation of biomolecules such as lipids, proteins, and DNA, which can further aggravate renal injury [174,175] (Table 1). Oxidative stress in CKD is also linked to impaired mitochondrial function and the heightened release of mitochondrial ROS, which contributes to the progression of renal injury and the development of atherosclerotic diseases [176]. Furthermore, patients with CKD may experience complications such as hypertension, atherosclerosis, inflammation, and anemia, which are associated with increased oxidative stress [177].

Drug-induced AKI typically manifests as two distinct patterns of renal injury: acute tubular necrosis (ATN) and acute interstitial nephritis (AIN). While AIN results from medications that trigger an allergic reaction, ATN arises from direct toxicity to tubular epithelial cells [178]. The development and progression of ATN are influenced by various cellular mechanisms, with oxidative stress being a critical factor that can trigger an inflammatory response through the release of pro-inflammatory cytokines and the accumulation of inflammatory cells in tissues. Specifically, oxidative stress can activate signaling pathways that contribute to the development of ATN by inducing mitochondrial dysfunction, lysosomal hydrolase inhibition, phospholipid damage, and increased intracellular calcium concentration. This leads to the overproduction of ROS and the depletion or inactivation of cellular antioxidants such as glutathione, which further exacerbate renal tubular cell death [179]. The resulting inflammation can contribute to the pathogenesis of ATN and its progression to AKI. These processes have been adequately described for drugs such as cisplatin and aminoglycosides, which can induce renal injury through mechanisms that are not related to their systemic pharmacological effects [180,181,182]. Instead, the accumulation of these drugs in proximal tubular cells promotes oxidative stress and mitochondrial damage, leading to apoptosis and necrosis. This process can trigger the development of fibrosis and inflammation, which further promotes the production of ROS and pro-inflammatory cytokines. As a result, a vicious cycle is created that is extremely detrimental to the kidney [179].

At this point, it is plausible to consider that other drugs, including cardiovascular medications, could induce renal injury through similar mechanisms, although the literature on the topic is limited. As extensively described in the preceding paragraphs, studies have suggested a potential involvement of ROS in renal injury that is mediated by statins and PPIs (Figure 2) (Table 1). It has also been hypothesized that CIN may involve excessive ROS production as one of its mechanisms of renal injury [166,183,184,185]. Conversely, drugs that do not cause renal damage, such as beta-blockers, have been widely recognized for their antioxidant properties, as can be deduced by the presence of OH groups in the molecule (Figure 1). This may help explain beta-blockers’ protective role for both the kidneys and the cardiovascular system [186,187,188,189].

Preclinical studies have demonstrated the significant and promising nephroprotective activity of multiple antioxidants, especially those derived from natural food sources [190], indicating their potential as effective sources of nephroprotective agents [191,192]. Therefore, targeting oxidative stress may offer a promising strategy to prevent or treat drug-induced nephrotoxicity and its associated morbidity and mortality. However, the current knowledge of the mechanisms and biomarkers of oxidative stress in drug-induced nephrotoxicity is still limited and requires further investigation. Moreover, the clinical efficacy and safety of antioxidant interventions in kidney disease are not well established and need to be evaluated in large-scale randomized controlled trials. Some of the future perspectives of oxidative stress and drug-induced nephrotoxicity include:developing novel and specific biomarkers of oxidative stress and drug-induced nephrotoxicity that can reflect the degree and location of renal injury, predict the risk and outcome of drug-induced nephrotoxicity, monitor the response to treatment, and guide personalized therapy [10];identifying novel and effective antioxidants that can target specific sources or pathways of oxidative stress and drug-induced nephrotoxicity, modulate redox signaling, protect renal cells and tissues from oxidative damage, and preserve or restore renal function [192];designing personalized and precise antioxidant therapy based on individual characteristics and needs, such as genetic background, epigenetic modifications, comorbidities, co-administered drugs, environmental factors, and oxidative stress status;exploring the potential synergistic or additive effects of combining antioxidant therapy with other therapeutic modalities, such beta-blockers [193];assessing the long-term benefits and risks of antioxidant therapy for drug-induced nephrotoxicity on various clinical outcomes, such as renal function preservation, cardiovascular protection, quality of life improvement, and survival extension.

## 5. Conclusions

Drug-induced nephrotoxicity is a serious complication that can affect the prognosis and quality of life of patients treated with various medications and diagnostic agents. The mechanisms of drug-induced nephrotoxicity vary depending on the type and class of the drug, the dose and duration of exposure, and the patient’s characteristics and comorbidities. Analyzing the structural characteristics of the various drugs, it is evident that they are all small molecules; therefore, it is entirely plausible to hypothesize that, especially in the presence of high dosages, these drugs can bind in a non-specific manner to other molecules different from their target, which could be the reason for the secondary effects of cytotoxicity. The prevention and management of drug-induced nephrotoxicity requires a comprehensive approach that includes several steps: first, a general knowledge of the mechanisms of drug-induced nephrotoxicity along with an understanding of the patient and drug-related risk factors that need to be corrected; then, assessment of baseline renal function before starting therapy, which involves the possible adjustment of drug dosage and avoidance of nephrotoxic drug combinations. Targeting oxidative stress could be a potential future solution for drug-induced nephrotoxicity, although further studies are needed.

## Figures and Tables

**Figure 1 metabolites-15-00191-f001:**
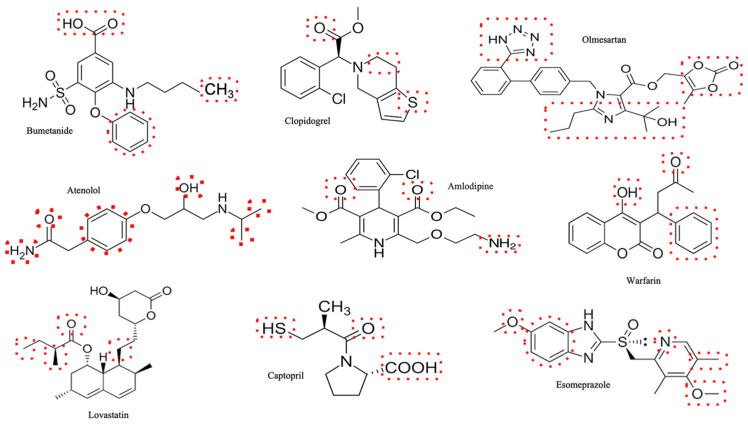
Schematic representation of the chemical structure of each different class of cardiovascular drugs. Chemical structures of Bumetanide (diuretics), clopidogrel (antiplatelet agents—APAs), Olmesartan (angiotensin receptor blockers—ARBs), atenolol (beta-blockers), amlodipine (calcium channel blockers—CCBs), warfarin (anticoagulants—ACs), Lovastatin (statins), Captopril (angiotensin-converting enzyme inhibitors—ACEIs), and Esomeprazole (proton-pump inhibitors—PPIs) are depicted. The dashed boxes highlight the parts of the molecule that interact with its target.

**Figure 2 metabolites-15-00191-f002:**
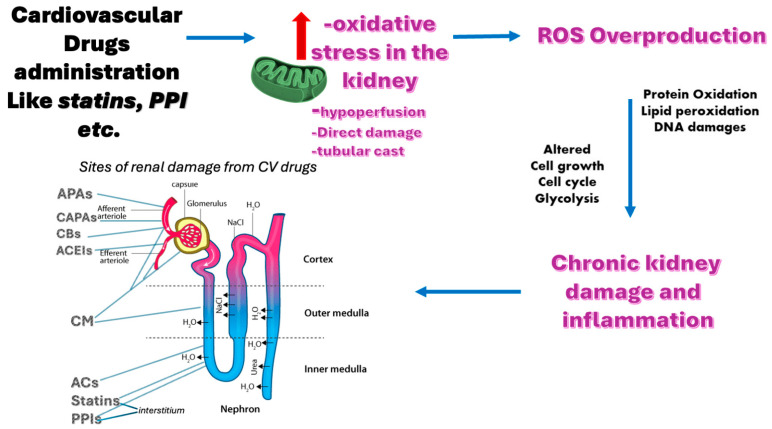
Renal toxicity of cardiovascular drugs. Cardiovascular drug administration could potentially cause renal damage with several mechanisms as indicated (top middle). However, a dominant recognized damaging mechanism is the enhancement of the oxidative stress in the renal tubules and interstitium; this leads to mitochondrial imbalance and ROS overproduction, which in turn can significantly alter cellular metabolism, facilitating a chronic inflammation state and ultimately leading to kidney disease. Specific sites of renal damage from CV drugs are also shown (bottom left). ACEIs = angiotensin-converting enzyme inhibitors; CCBs = calcium channel blockers; BBs = beta-blockers; APAs = antiplatelet agents; ACs = anticoagulants; PPIs = proton-pump inhibitors; CM = contrast medium; ROS = reactive oxygen species.

**Table 1 metabolites-15-00191-t001:** Summary of the nephrotoxic effects of cardiovascular drugs.

	Glomerulus	Tubules	Other Renal Toxic Effects	Factors That Enhance Toxicity
Diuretics (loop)	Reduction of RBF and GF	Obstruction (Tamm–Horsfall protein formation)	Paradoxical stimulation of RAAS	ACEIs/NSAIDs
ACEIs and ARBs	Reduced IG-P (dilation Eart)	-	-	NSAIDs and diuretics (triple whammy)
CCBs	Increase IG-P (dilation Aart) (more proteinuria)	-	-	CYP3A4 inhibitors (Clarithromycin)
BBs	Decrease in GFR (mild)	-	-	-
APAs	Decrease in GFR for intense vasoconstriction (PGEi)	AIN	-	Dehydration, HF, LC, sepsis, diuretics, and ACEIs; statins
ACs	Occlusive casts Bowman space	Occlusive cast	-	Dehydration
Statins	-	TIN	Suppression Q10	Rhabdomyolysis
PPIs	-	TIN	Hypomagnesemia	Enteric infection
CM	Vasospasm	Direct tubular cell damage	-	Hypotension, microembolism, and bleeding; metformin

ACEIs and ARBs = angiotensin-converting enzyme inhibitors and angiotensin receptor blockers; CCBs = calcium channel blockers; BBs = beta-blockers; APAs = antiplatelet agents; ACs = anticoagulants; PPIs = proton-pump inhibitors; tox = toxicity; NSAIDs = non-steroidal anti-inflammatory drugs; PGEi = prostaglandins inhibition; CM = contrast media; RBF = renal blood flow; GFR = glomerular filtration rate; HF = heart failure; LC = liver cirrhosis; Aart = afferent arteriole; Eart = efferent arteriole; RP = peripheral resistance; TIN = tubular-interstitial nephritis; AIN = acute interstitial nephritis; IG-P = intraglomerular pressure; RAAS = renin–angiotensin–aldosterone system.

## Data Availability

No new data were created or analyzed in this study.

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
