# Peer review of "Kidney Toxicity of Drugs for the Heart: An Updated Perspective"

_metabolites, 2025, doi:10.3390/metabo15030191_

Round 1
Reviewer 1 Report
Comments and Suggestions for Authors
In the current manuscript in Metabolites (Manuscript ID: metabolites-3448131), Carlo Caiati et al. have reviewed on the impact of cardiovascular drugs on the kidney toxicity. These drugs are predominantly used for the treatment of cardiovascular disorders and same are causing adverse impact and leading to chronic kidney diseases. Overall, this is a well-compiled review article with important insights that summarize the existing information on the renal toxicity due to some common cardiovascular drugs.
This manuscript could attract substantial interest from readers of the journal Metabolites.
However, I do recommend a major revision, and addressing the following comments could further improve the manuscript:
Comments:
1. Title: Kidney toxicity of the drugs for the heart. An updated perspective.
Comment: Kindly reframe the title. The term “drugs for the heart..” doesn’t sound scientific.
2. Line 28: “All the waste products in particular can reach a high concentration in the kidneys for 3 main reasons.”
Comment: This statement is not clear to the reader. Please reframe it to explain what author means? Further, spelling mistake in Line 29. The three points explained further also are not clear in the way they are explained.
I would request the authors to rephrase the first paragraph of the introduction to provide more engaging points to drive the thoughts of the review.
3. Line 43: “For all these reasons the kidney is a major target organ for many drugs,…”
Comment: Reframe this sentence, as kidney is not a major target, rather it is primarily easily affected by drugs taken by a pateint for other clinical conditions.
4. Line 53: “Cardiovascular drugs are among the most frequently prescribed medications worldwide,…..”
Comment: Provide reference for this sentence.
5. Line 105-108: “The renal tubules, especially the proximal tubule cells, are exposed to drugs that undergo concentration and re-absorption in the tubules, and therefore are highly susceptible to drug toxicity [14].”
Comment: Rephrase this sentence as it is not able to convey a meaningful scientific expression to the readers.
6. Line 88-89: “Glomerular hemodynamics refers to the blood flow and pressure within the glomerulus, ”
Comment: Either rephrase it or break it in two sentences. It seems like author is saying that Glomerular hemodynamics is the network of capillaries that filters blood and forms urine, which is not.
7. Line 105-107: “The renal tubules, especially the proximal tubule cells, are exposed to drugs that undergo concentration and reabsorption in the tubules, and therefore are highly susceptible to drug toxicity.”
Comment: Reframe it as it is written poorly.
8. Line 114: “They disrupt cell membrane of the tubular cells but also mitochondria organelles.”
Comment: Replace ‘but’ with ‘and’.
9. Line 125-126: “Inflammation that impairs normal kidney functions and causes toxicity includes glomerulonephritis, acute and chronic interstitial nephritis, or vasculitis [22].”
Comment: Reframe it.
10. Line 223-224: “We recommend cautious use of diuretics and ACE inhibitors in patients especially those with underlying kidney disease.”
Comment: Recommendation on using any drugs for clinical conditions can be done by clinical physicians. In a review like this, authors can have an opinion on such perspective. This manuscript is based on already published data, so authors may refer to any such recommendation made by any original studies. Kindly reframe it.
11. Line 180: Renal effects of cardiovascular drugs
Comments: It is a vague title and coveys otherwise. Be more precise.
12. Line 452 455: “They work by reducing the levels of low-density lipoprotein (LDL), which is a major contributor to atherosclerosis and coronary heart disease total cholesterol, and triglycerides, while increasing high-density lipoprotein (HDL) concentrations.”
Comment: Frame small and meaningful sentences.
13. Line 464-466: “The statins more directing impacting the kidneys are the hydrophilic ones like pravastatin and rosuvastain especially at high dosage.”
Comment: Grammatical error. Reframe it.
14. Line 469-473: “One possible mechanism involves the potential of statins, especially at high dosage, to cause rhabdomyolysis, that is a condition characterized by the breakdown of skeletal muscle, which can lead to the release of sarcoplasmic proteins and electrolytes, that can lead to AKI and other potentially life-threatening complications, such as hyperkaliemia and cardiac arrhythmias.”
Comment: Refrain from framing long sentences with several information. It is like confusing and lose its informative value by the time one reach to the end.
15. Line 509: “..and Helicobacter pylori infection by..”
Comment: Bacterial name should be in italics. Similar comment for line 524.
16. Line 575: Role of oxidative stress in drug-induced nephrotoxicity
Comment: The title of the review is on the drugs (given for cardiovascular diseases) that can cause kidney complications. Oxidative stress is a consequence of drugs usage or disease conditions. In fact, cardiovascular complications itself result in oxidative stress that had adverse effects on kidney. Further, various uremic toxins are well established to cause oxidative stress via various mechanism. How would you justify discussing role of oxidative stress in kidney diseases? Kindly elaborate.
17. Line 661-665: “The prevention and management of drug-induced nephrotoxicity require a comprehensive approach that involves knowledge of the mechanisms of drug-induced nephrotoxicity, understanding of the patient and drug-related risk factors, and therapeutic intervention by correcting risk factors, assessing baseline renal function before initiation of therapy, adjusting the drug dosage and avoiding use of nephrotoxic drug combinations.
Comment: Statement is too long. Frame small and meaningful sentences.
18. Overall Comment:
a. This review needs major revision.
b. The English needs improvement.
c. This review needs tabular content to sum up the category of drugs and various mechanisms explained. A table can summarize the information in more effective way.
Authors can be creative to depict some pictorial presentation of the facts mentioned, to be more engaging and illustrative. Just one figure with molecular structure of some drugs is not sufficient.
Comments on the Quality of English LanguageEnglish language can be improved to more clearly express. Try to re-frame the smaller meaning full sentences.
Author Response
Reviewer 1
In the current manuscript in Metabolites (Manuscript ID: metabolites-3448131), Carlo Caiati et al. have reviewed on the impact of cardiovascular drugs on the kidney toxicity. These drugs are predominantly used for the treatment of cardiovascular disorders and same are causing adverse impact and leading to chronic kidney diseases. Overall, this is a well-compiled review article with important insights that summarize the existing information on the renal toxicity due to some common cardiovascular drugs.
This manuscript could attract substantial interest from readers of the journal Metabolites.
However, I do recommend a major revision, and addressing the following comments could further improve the manuscript:
Comments:
- Title: Kidney toxicity of the drugs for the heart. An updated perspective.
Comment: Kindly reframe the title. The term “drugs for the heart.” doesn’t sound scientific.
Answwer: Thanks for this observation. I know that this expression doesn’t sound very scientific but “Drugs for the heart” is the title of the the most famous books of Cardiovascular drugs: “Drugs for the Heart” by Opie and Gersh. So the title has been formulated using the same book terms as if the paper was an integration of the book itself. I would prefer to leave the expression like it is. However cardiovascular drugs is ok.
- Line 28: “All the waste products in particular can reach a high concentration in the kidneys for 3 main reasons.”
Comment: This statement is not clear to the reader. Please reframe it to explain what author means? Further, spelling mistake in Line 29. The three points explained further also are not clear in the way they are explained.
I would request the authors to rephrase the first paragraph of the introduction to provide more engaging points to drive the thoughts of the review.
Answwer: Thanks for these comments. Accordingly, I rephrased this paragraph (in revision mode in the paper, line 45)
- Line 43: “For all these reasons the kidney is a major target organ for many drugs,…”
Comment: Reframe this sentence, as kidney is not a major target, rather it is primarily easily affected by drugs taken by a pateint for other clinical conditions.
Answwer: Thanks for this comments. I totally agree so I rephrased accordingly: (in revision mode in the paper, Line 45).
- Line 53: “Cardiovascular drugs are among the most frequently prescribed medications worldwide,…..”
Comment: Provide reference for this sentence.
Answwer: done (line 66, ref 8)
- Line 105-108: “The renal tubules, especially the proximal tubule cells, are exposed to drugs that undergo concentration and re-absorption in the tubules, and therefore are highly susceptible to drug toxicity [14].”
Comment: Rephrase this sentence as it is not able to convey a meaningful scientific expression to the readers.
Answwer: Many thanks for this appropriate suggestion; accordingly I completely rephrased the sentence (in revision mode in the paper, Line 128).
- Line 88-89: “Glomerular hemodynamics refers to the blood flow and pressure within the glomerulus, ”
Comment: Either rephrase it or break it in two sentences. It seems like author is saying that Glomerular hemodynamics is the network of capillaries that filters blood and forms urine, which is not.
Answwer: Very grateful for this appropriate suggestion; accordingly I completely rephrased the sentence (in revision mode in the paper, Line 109).
- Line 105-107: “The renal tubules, especially the proximal tubule cells, are exposed to drugs that undergo concentration and reabsorption in the tubules, and therefore are highly susceptible to drug toxicity.”
Comment: Reframe it as it is written poorly.
Answwer: already done (see comments 5)
- Line 114: “They disrupt cell membrane of the tubular cells but also mitochondria organelles.”
Comment: Replace ‘but’ with ‘and’.
Answwer: done (line 137)
- Line 125-126: “Inflammation that impairs normal kidney functions and causes toxicity includes glomerulonephritis, acute and chronic interstitial nephritis, or vasculitis [22].”
Comment: Reframe it.
Answwer: Many thanks again for this appropriate suggestion; that was done (in revision mode in the paper line 146)
- Line 223-224: “We recommend cautious use of diuretics and ACE inhibitors in patients especially those with underlying kidney disease.”
Comment: Recommendation on using any drugs for clinical conditions can be done by clinical physicians. In a review like this, authors can have an opinion on such perspective. This manuscript is based on already published data, so authors may refer to any such recommendation made by any original studies. Kindly reframe it.
Answwer: we totally agree; accordingly we have reframed that sentence (line 291)
- Line 180: Renal effects of cardiovascular drugs
Comments: It is a vague title and coveys otherwise. Be more precise.
Answwer: we totally concord; we have reframed, being more specific (line 226)
- Line 452 455: “They work by reducing the levels of low-density lipoprotein (LDL), which is a major contributor to atherosclerosis and coronary heart disease total cholesterol, and triglycerides, while increasing high-density lipoprotein (HDL) concentrations.”
Comment: Frame small and meaningful sentences.
Answwer: we agree, consequently we have reframed the paragraph (line 532)
- Line 464-466: “The statins more directing impacting the kidneys are the hydrophilic ones like pravastatin and rosuvastatin especially at high dosage.”
Comment: Grammatical error. Reframe it.
Answwer: done (line 541)
- Line 469-473: “One possible mechanism involves the potential of statins, especially at high dosage, to cause rhabdomyolysis, that is a condition characterized by the breakdown of skeletal muscle, which can lead to the release of sarcoplasmic proteins and electrolytes, that can lead to AKI and other potentially life-threatening complications, such as hyperkaliemia and cardiac arrhythmias.”
Comment: Refrain from framing long sentences with several information. It is like confusing and lose its informative value by the time one reach to the end.
Answwer: we totally agree; so we have rephrased this long sentence (line 561)
- Line 509: “..and Helicobacter pylori infection by..”
Comment: Bacterial name should be in italics. Similar comment for line 524.
Answwer: thanks; we have used italics (line 600 and 642)
- Line 575: Role of oxidative stress in drug-induced nephrotoxicity
Comment: The title of the review is on the drugs (given for cardiovascular diseases) that can cause kidney complications. Oxidative stress is a consequence of drugs usage or disease conditions. In fact, cardiovascular complications itself result in oxidative stress that had adverse effects on kidney. Further, various uremic toxins are well established to cause oxidative stress via various mechanism. How would you justify discussing role of oxidative stress in kidney diseases? Kindly elaborate.
Answwer: Thanks for this comment; a special covering has been made regarding this mechanism since we think that it is very relevant to the explanation of cardiovascular drugs toxicity; also, because the kidneys are highly susceptible to damage caused by oxidative stress due to the intense oxidative activity of their mitochondria, as reported in text. So, any factor like drugs, that further increase oxidative stress can be very damaging for the kidneys.
- Line 661-665: “The prevention and management of drug-induced nephrotoxicity require a comprehensive approach that involves knowledge of the mechanisms of drug-induced nephrotoxicity, understanding of the patient and drug-related risk factors, and therapeutic intervention by correcting risk factors, assessing baseline renal function before initiation of therapy, adjusting the drug dosage and avoiding use of nephrotoxic drug combinations.
Comment: Statement is too long. Frame small and meaningful sentences.
Answwer: done (line 814)
- Overall Comment:
- This review needs major revision.
- The English needs improvement.
Answwer: we have improved the English
- This review needs tabular content to sum up the category of drugs and various mechanisms explained. A table can summarize the information in more effective way.
Answwer: we have added at the end a table (Table) that summarizes the renal toxic effects of the cardiovascular drugs.
Authors can be creative to depict some pictorial presentation of the facts mentioned, to be more engaging and illustrative. Just one figure with molecular structure of some drugs is not sufficient.
Answwer: as required, we have added a Figure (Figure 2) that pictorially describes the content of the review.
Comments on the Quality of English Language
English language can be improved to more clearly express. Try to re-frame the smaller meaning full sentences.
Answwer: we have attempted to do that
Thanks very much for your very useful and constructive suggestions.
Submission Date
11 January 2025
Date of this review
20 Jan 2025 02:06:34
Reviewer 2 Report
Comments and Suggestions for Authors The review entitled: ''Kidney toxicity of the drugs for the heart. An updated perspective'' is high quality work, with comprehensive background and number of supported literature. The quality of English is good, text is easy to understand. The conclusions are clear and comprehensive. My recommendation is a minor change of title, i.e. Kidney toxicity of cardiac drugs. An updated perspective The graphical abstact introducing will be beneficial. To sum up, excellent work. Congratulations.
2
perspective
Author Response
The review entitled: ''Kidney toxicity of the drugs for the heart. An updated perspective'' is high quality work, with comprehensive background and number of supported literature. The quality of English is good, text is easy to understand. The conclusions are clear and comprehensive. My recommendation is a minor change of title, i.e. Kidney toxicity of cardiac drugs. An updated perspective The graphical abstact introducing will be beneficial. To sum up, excellent work. Congratulations.
Answwer: Thanks very much for your appreciation; regarding the title as already said to the first reviwer, we know that “drugs for the heart” expression sounds not very scientific but “Drugs for the heart” is the title of the the most famous books of Cardiovascular drugs: “Drugs for the Heart” by Opie and Gersh. So the title has been formulated using the same book terms as if the paper was an integration of the book itself. I would prefer to leave the expression like it is. However cardiovascular drugs is ok.
Regarding the second point, we have added a graphical abstract as you and the first reviewer have suggested (Figure 2).
Reviewer 3 Report
Comments and Suggestions for Authors
i have reviewed the manuscript entitled "Kidney toxicity of the drugs for the heart. An updated perspective". The article is an extended, updated review about drugs used to treat cardiac diseases that have a potential kidney involvement in producing AKI.
The article is well structured and very comprehensive and details one by one the drugs category. The references are appropriate, extensive and relevant for this research. Congratulations to the authors for this very good manuscript!
Author Response
Thanks very much for you appreciation.
Round 2
Reviewer 1 Report
Comments and Suggestions for Authors
I am happy how author's have taken care of all the queries and have definitely did a job job on improving the article structure, illustration and presentation.
I do recommend the article for publication.